# Towards establishing a fungal economics spectrum in soil saprobic fungi

Tessa Camenzind [1,2] ✉, Carlos A. Aguilar-Trigueros[3,4], Stefan Hempel[1,2], Anika Lehmann [1,2], Milos Bielcik[1,2], Diana R. Andrade-Linares[5], Joana Bergmann [6], Jeane dela Cruz [1,2], Jessie Gawronski[1,2], Polina Golubeva[1,2], Heike Haslwimmer[7], Linda Lartey[1,2], Eva Leifheit[1,2], Stefanie Maaß[1,2], Sven Marhan[7], Liliana Pinek[1,2], Jeff R. Powell[3], Julien Roy[1,2], Stavros D. Veresoglou[8], Dongwei Wang[1,2], Anja Wulf[1,2], Weishuang Zheng[9] & Matthias C. Rillig [1,2]

Trait-based frameworks are promising tools to understand the functional consequences of community shifts in response to environmental change. The applicability of these tools to soil microbes is limited by a lack of functional trait data and a focus on categorical traits. To address this gap for an important group of soil microorganisms, we identify trade-offs underlying a fungal economics spectrum based on a large trait collection in 28 saprobic fungal isolates, derived from a common grassland soil and grown in culture plates. In this dataset, ecologically relevant trait variation is best captured by a three-dimensional fungal economics space. The primary explanatory axis represents a dense-fast continuum, resembling dominant life-history trade-offs in other taxa. A second significant axis reflects mycelial flexibility, and a third one carbon acquisition traits. All three axes correlate with traits involved in soil carbon cycling. Since stress tolerance and fundamental niche gradients are primarily related to the dense-fast continuum, traits of the 2nd (carbon-use efficiency) and especially the 3rd (decomposition) orthogonal axes are independent of tested environmental stressors. These findings suggest a fungal economics space which can now be tested at broader scales.

Soils are exposed to multiple anthropogenic pressures and changing environmental conditions that threaten microbial functions essential for biogeochemical processes maintaining soil sustainability[1]. Soil microbial communities are affected by environmental change—with a plethora of sequencing studies revealing shifts in microbial community structure following changes in environmental conditions[2]. Yet, the much needed projection of community shifts onto microbial functions lags behind[3,4]. Achieving such functional predictability and mechanistic understanding in soil microbial ecology represents one of the frontiers in this field. In particular, predicting the microbial cycling of soil organic carbon (C) requires in-depth knowledge about microbial functions[5,6]. There is an ongoing debate whether and how

[1]Institute of Biology, Freie Universität Berlin, Altensteinstr. 6, 14195 Berlin, Germany. [2]Berlin-Brandenburg Institute of Advanced Biodiversity Research (BBIB), Berlin, Germany. [3]Hawkesbury Institute for the Environment, Western Sydney University, Penrith, NSW 2751, Australia. [4]Department of Biological and Environmental Science, University of Jyväskylä, P.O. Box 35, 40014 Jyväskylä, Finland. [5]Research Unit Comparative Microbiome Analysis, Helmholtz Zentrum München, Ingolstaedter Landstraße 1, 85764 Neuherberg, Germany. [6]Leibniz Centre for Agricultural Landscape Research (ZALF), 15374 Müncheberg, Germany. [7]Institute of Soil Science and Land Evaluation, Soil Biology department, University of Hohenheim, Emil-Wolff-Str. 27, 70599 Stuttgart, Germany. [8]State Key Laboratory of Biocontrol, School of Ecology, Sun Yat-sen University, Shenzhen 518107, China. [9]Marine Institute for Bioresources and Environment, Peking University Shenzhen Institute, Shenzhen 518057, China. ✉e-mail: tessa.camenzind@fu-berlin.de

predictability in complex soil microbial communities can be achieved[7], which is especially relevant in the context of rapidly advancing soil biogeochemical models[5]. A most promising tool in this context is given by trait-based frameworks, which directly couple community composition to function[8]. These frameworks proved to be powerful in global models implementing plant traits, and there is considerable effort now to establish such frameworks in soil microbiology[3,9].

The strength of functional trait-based approaches lies in the simple concept of constrained resource allocation possibilities—organisms cannot be good at everything[10]. Consequently, trade-offs allow to predict that species with certain trait expressions will not express functions at the other end of a trait continuum[11]. Frameworks like the widely applied Grime´s C-S-R (competitor-stress tolerant-ruderal) triangle expand on this simple idea and sort species into respective life history strategies[12,13]. This concept has been recently applied to microbial soil C cycling functions - the Y-A-S (yield-acquisition-stress tolerant) or similar C-S-O (competitors-stress tolerators-opportunists) framework[6,14]. These simple frameworks are regarded as promising tools to implement microbial traits in soil C models[5,15], since they directly connect environmental stressors to C cycling properties: If traits indicative of stress tolerance trade-off with C acquisition rates, this enables the prediction of a decrease in C mineralization in response to respective stressors at the community level (stress is defined here as an external constraint limiting fungal growth[12], anthropogenic stress refers to external growth constraints induced or intensified by human activities). In practice, though, these simple categories fail to capture the continuous nature of trait expressions along independent dimensions, are highly subjective in their interpretation and lack objective quantifiable trait matrices. Due to these

drawbacks, in plant ecology these categorical frameworks have been replaced a while ago by the idea of an economics spectrum[16,17]. Based on relatively few traits, plants can be directly sorted on a main trait axis along the spectrum of conservative to acquisitive traits, also referred to as the slow-fast continuum[18]. The main axes of the plant economics spectrum explain life-history strategies[19], relate to environmental parameters[20] and are relevant predictors in global models[21]. Following this powerful approach, it has been suggested to similarly transition to a trait continuum in microbial ecology rather than focusing on inconclusive functional groups[22].

Microbial ecology clearly lags behind in trait-based ecology. This is no surprise given the tremendous diversity within soil microbial communities and the challenging methodological impracticalities preventing in vivo measurements of many traits of individual microbes. Consequently, experimental evidence on functional (multivariate) trait axes remains scarce[23–26]. Rapidly evolving soil metagenomics allows the analyses of microbial traits at larger scales[27]. However, genomic information only provides a glimpse on the potential of an organism, not its actual phenotype[22]. More importantly, these data provide little information on primary trade-offs in resource investment in microorganisms. To make genomic data applicable, such functional trade-offs must be established first in the organisms itself. Advanced experimental insights into a broader spectrum of microbial functional traits and their integration in an environmental context are needed[3,9]. At this stage, existing trait frameworks on soil microbes are only based on conceptual considerations of microbial physiology, lacking data to support them[6,28–30]. Major concepts are simply derived from macro-ecological theory, which may explain their limitations to predict soil C dynamics[31,32]. Thus, in a first step we may

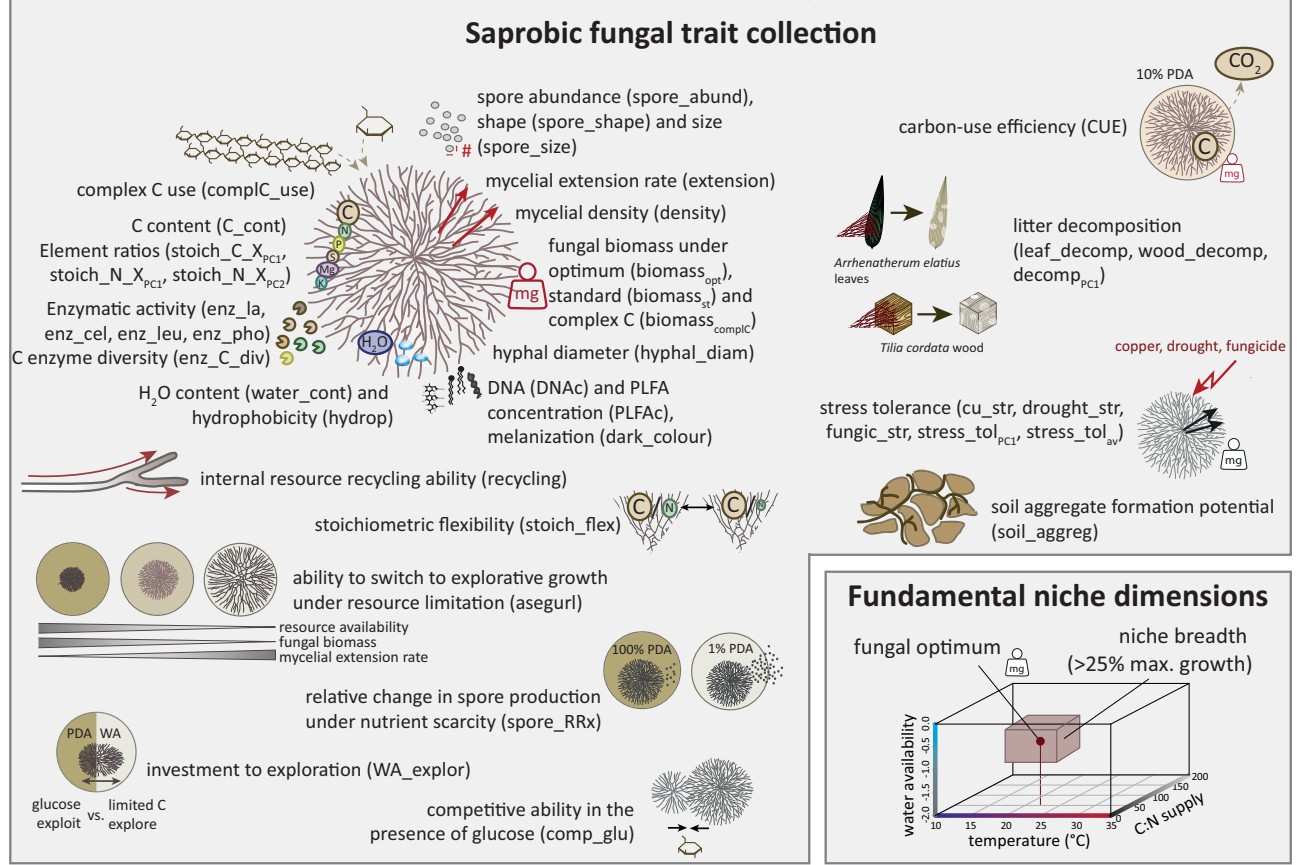

**Fig. 1 | Visual illustration of the comprehensive fungal trait collection analyzed in this study.** Trait abbreviations used throughout the manuscript are added in brackets (detailed explanation of traits can be found in Table S1). Some of the trait data used here represent functional traits not previously measured in saprobic fungi, others have been published in previous studies (see Table S2). C carbon, N nitrogen, P phosphorus, S sulfur, Mg magnesium, K potassium, X nutrient, PC principal component axis, PLFA phospholipid fatty acids, PDA potato dextrose agar, WA water agar, Cu copper, mg milligram.

monitor correlations of functional traits in microbial isolates; in a way comparable with the establishment of main trait axes in small sets of plant species under common garden experiments that took place more than 20 years ago[16,33].

We here address these challenges and provide insights into relevant functional traits in saprobic soil fungi. Saprobic fungi in soil represent a highly diverse microbial guild that drives C and nutrient turnover, contributes large portions of soil microbial biomass[34,35] and activity[36] and plays a major role in soil C sequestration[37,38]. Although this group exhibits a common phenotype including filamentous growth (though not exclusively) and osmotrophy, fungal natural history records reveal a broad phylogenetic diversity of soil saprobic fungi, as well as a wide variety of growth forms. Such diverse growth forms are indicative of distinct resource allocation strategies that may have significant impacts on C and nutrient cycling dynamics currently unrecognized in microbial ecology. An economics spectrum for saprobic fungi would represent a major milestone in microbial ecology. Relevant functional traits driving such a spectrum must capture the function and ecology of fungal mycelia developing in soil, characterized by exploration (exploring and connecting different resource patches) and exploitation (intensively exploiting resources with densely branched mycelia and enzymes produced)[39,40]. Saprobic fungi must invest resources to explore the heterogeneous soil environment, both with new hyphae formed as well as asexual spores produced, which enable fast dispersal and colonization of new resource patches[29,41]. In return, energy and nutrients become available for growth as simple sugars (and nutrients), or complex C sources demanding various levels of enzymatic capacity. In this respect, relevant functional traits include mycelial architecture and composition, mycelial strategies indicative of exploration vs. exploitation (including internal resource recycling and translocation within hyphae), asexual spore production, and resource uptake strategies[13,42,43]. While several authors have in theory suggested many traits fitting those criteria[44,45], in practice too few trait combinations have been measured to establish general trait axes in soil saprobic fungi[25,46,47].

Therefore, we made a first attempt to establish primary trait axes of a fungal economic spectrum/space in soil saprobic fungi. We included morphological, physiological and life-history variables from a phylogenetically diverse set of fungi isolated at spatial scales where they are likely to interact (Fig. 1; traits were assessed in vitro). Since saprobic soil fungi interact at a much smaller scale than plants, this set of 28 saprobic fungal isolates originating from the same grassland soil (Fig. 1, Table S3) provides an appropriate test group. Primary trade-offs in resource investment (e.g., acquisitive vs. conservative) should act on a local scale of species interactions, and ideally expand to larger scales[33]. The environmental relevance of the derived economics space was further tested by correlations with fundamental niche dimensions. In this study we tested whether (i) the primary fungal economics spectrum will reflect successional patterns observed in other organism groups, i.e., a slow-fast spectrum[18,48], (ii) fast growth strategies will be supported by efficient hyphal strategies and rapid asexual sporulation and (iii) the main axes of the fungal economics space will relate to stress tolerance and be a good predictor for fundamental niche spaces of fungal isolates. Although our trait measurements come from artificial lab conditions (experiments were conducted in petri dishes), we believe that they represent a meaningful first step to infer the resource allocation strategies of saprobic soil fungi.

## Results and discussion
### Functional traits forming the fungal economics spectrum
We assembled a large dataset to describe a fungal economics spectrum for 28 (potentially) co-occurring soil fungal isolates, including several traits that have been assessed for the first time in fungi (to the best of our knowledge; Fig. 1, Table S1). Despite the strength of this large trait collection, it would not be expedient to include every available trait, also due to the complex correlation structure among them (Fig. S1). Instead, functional traits capturing most variation and being ecologically significant in soil were to be selected. This has been a successful strategy in plants[16]; though even in plant ecology—despite extensive trait databases—it is a long-lasting ongoing endeavor to define these traits[18,49,50]. Here, we defined primary trait axes starting with a randomized principal component analysis (PCA) method. This approach revealed a consistent first principal component (PC) axis capturing a continuous spectrum of slow to fast growing fungal isolates (Fig. S2, S3). While slow fungi were characterized by dense highly-branched mycelia, fast fungal isolates showed high mycelial extension rates, i.e.,

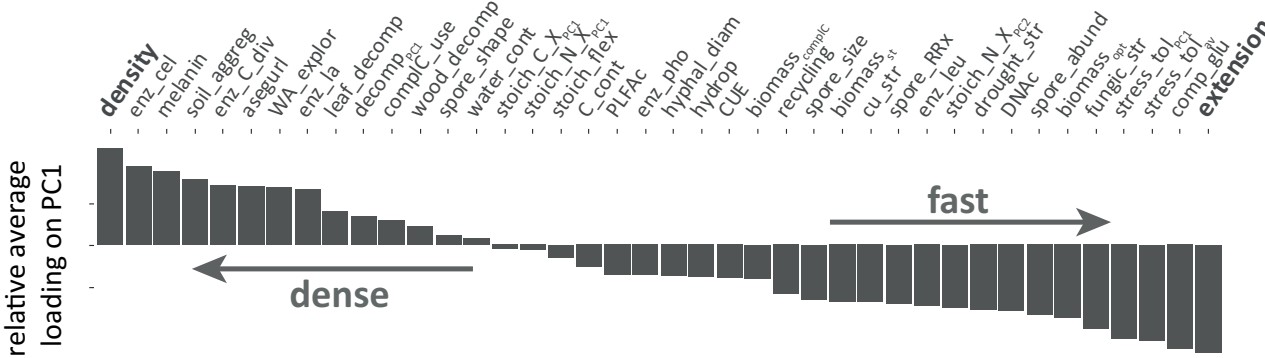

**Fig. 2 | Alignment of all fungal traits with the primary dense-fast spectrum in saprobic fungi based on a randomized principal component analysis (PCA) approach.** Bars display average loadings of individual functional traits on PC1 (principal component axis 1) based on 10,000 PCA repeats with random inclusions of 10 traits, in addition to extension and density. Abbreviations from left to right: enz_cel enzymatic activity of cellobiohydrolase, soil_aggreg soil aggregate formation ability, enz_C_div C enzyme diversity, asegurl ability to switch to explorative growth under resource limitations, WA_explor investment to exploration, enz_lac enzymatic activity of laccase, leaf_decomp leaf litter decomposition, decomp_PC1 1st PC axis of leaf and wood decomposition ability, complC_use complex carbon use, wood_decomp wood decomposition, water_cont water content, stoich_C_X_PC1 1st PC axis of stoichiometric C:X (nutrient) ratios, stoich_N_X_PC1 1st PC axis of stoichiometric N:X (nitrogen: nutrient) ratios, stoich_flex stoichiometric flexibility, C_cont fungal C content, PLFAc total PLFA concentration, enz_pho enzymatic activity of phosphatase, hyphal_diam hyphal diameter, hydrop mycelial hydrophobicity, CUE carbon-use efficiency, biomass_complC fungal biomass on complex C sources, biomass_st fungal biomass under standard conditions, cu_str stress tolerance to copper, spore_RRx relative change in spore production under nutrient scarcity, enz_leu enzymatic activity of leucine aminopeptidase, stoich_N_X_PC2 2nd PC axis of stoichiometric N:X (nitrogen: nutrient) ratios, drought_str stress tolerance to drought; DNAc: DNA concentration, spore_abund spore abundance, biomass_opt fungal biomass under optimal conditions, fungic_str stress tolerance to fungicide, stress_tol_PC1 1st PC axes of stress tolerance (copper, drought, fungicide), stress_tol_av average stress tolerance (copper, drought, fungicide), comp_glu competitive ability under glucose supply.

investment to rapid tip growth. This primary axis likely represents a physiological trade-off based on the simple nature of mycelial development (characterized by tip growth, branching and negative autotropism[39]), but also reflects the fungal strategies of exploitation versus exploration in soil. The correlation (PC loadings) of further functional traits with this 1st PC axis revealed an overall trait continuum from slow growing (dense) fungi with high melanin contents and larger enzymatic capacities to fast growing competitive fungi, characterized by high sporulation, yield, and stress tolerance (Fig. 2). The low average loadings of many traits on this 1st axis further indicated the significance of additional relevant functional trait axes.

In order to analyze the distinct patterns of a complete fungal economics space in more detail, we then selected traits with high functional and ecological significance for saprobic fungal growth in soil (Fig. 3, see methods section). The prior reduction of trait variables did not reduce the number of PC axes being significant (Fig. S4), making us confident that selected traits captured the fungal economics space well. In this final PCA (Fig. S5), the loadings of functional traits on PC axes clustered as distinct ecological groups of trait spectra (Fig. 3a). The first PC axis−representing the largest variation present in the data (29%)−roughly reflected the dense-fast continuum described above (Figs. 2 and 3a, S5). Additionally, two further axes were statistically significant: The second PC axis was mainly formed by traits

related to mycelial strategies, while the third axes showed high loadings of C acquisition traits (Fig. 3a). These patterns led to distinct ecological clusters of trait variables (Fig. 3a), coinciding with stronger correlations among these trait variables (Fig. S1).

In order to maximize the interpretation of functional trait axes representative of the fungal economics space we implemented a varimax rotation (a simple rotation maintaining the orthogonal (linearly uncorrelated) structure of PC axes; Fig. 3, Fig. S5). This led to three RC (rotated component) axes, each reflecting distinct ecological trait spectra. The first axis (RC1 dense-fast) represented a clear slow-fast continuum (Fig. 3b), here referred to as the dense-fast spectrum: A continuum of slow growing (dense and highly branched, i.e., exploitative) fungi to fast growing, competitive, extensively sporulating fungal isolates (explorative)[39]. The dense side of the spectrum was further characterized by high melanin contents, as well as an ability to switch to explorative growth under resource limitation which to our knowledge is a novel trait (we refer to it with the abbreviation 'asegurl'; Fig. S6). The fast side coincided with high stress tolerance, here defined by a low growth reduction in response to different stressors (i.e., copper, drought and fungicides). A second axis (RC2 flexibility) emerged from traits representing mycelial growth strategies, i.e., internal resource recycling and flexible growth, as well as stoichiometric flexibility in response to resource limitations. These variables

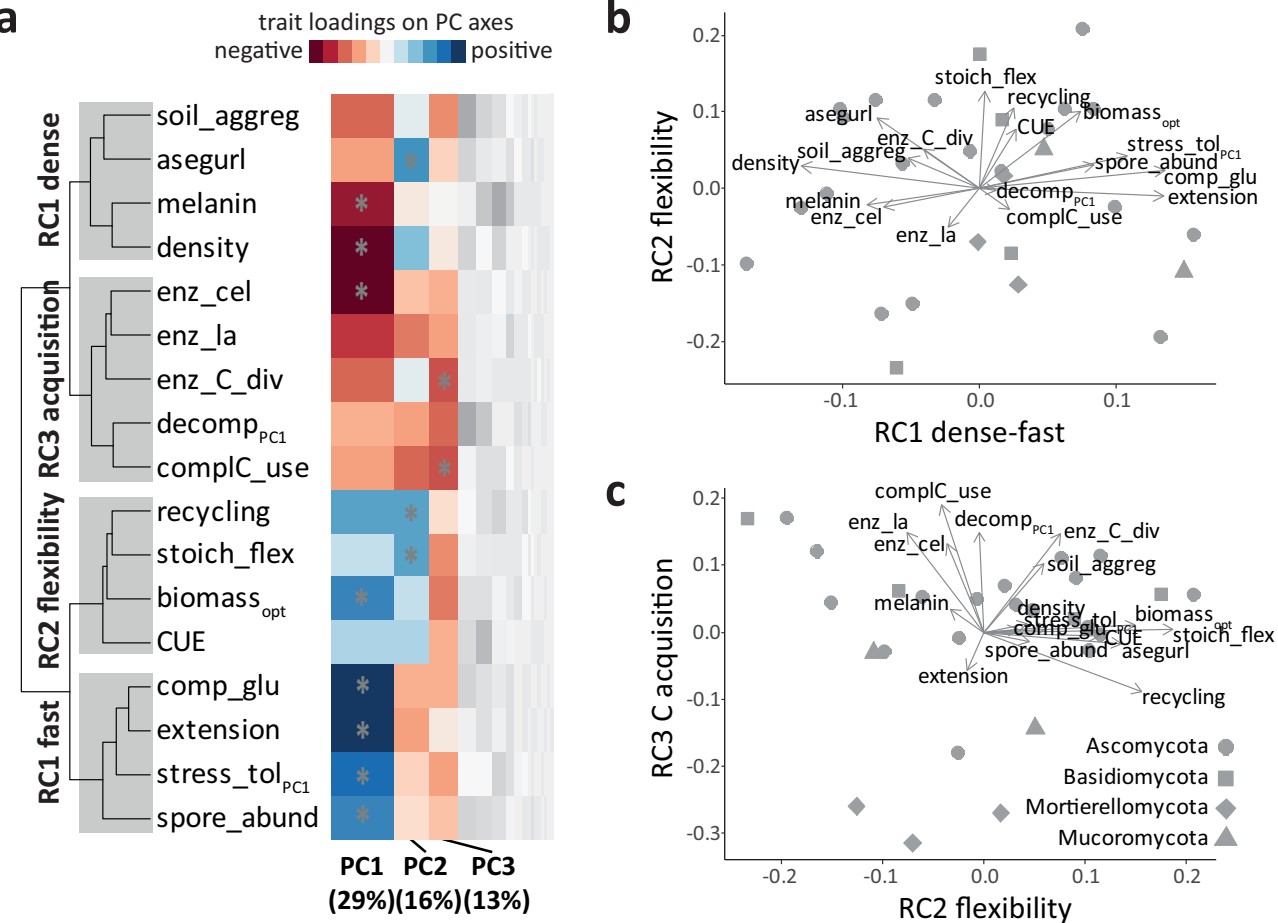

**Fig. 3 | Visualization of the main three-dimensional fungal economics space in saprobic fungal isolates based on functional traits. a** Heatmap visualization of the main principal component analysis (PCA, Fig. S5), with loadings of individual traits on PC axes being displayed, as well as their groupings (shown as correlogram based on hierarchical cluster analyses of loadings). The column width corresponds to the eigenvalues of PC axes (insignificant axes are shown in gray scales), asterisks indicate significant loadings of traits on respective axes (based on PCAtest[93]). Red colors indicate negative, blue colors positive loadings, with color intensity referring

to respective loading values. The clustering groups of traits reflect their loadings on varimax rotated components (RC), highlighted by gray squares. Fungal economics space defined by PCA followed by varimax rotation, showing the rotated components (RC) RC1 and 2 (**b**) and RC2 and 3 (**c**). Arrows represent eigenvectors of traits on RC axes, dots individual isolates (shapes of dots indicate phylogenetic placement). For details on functional traits see Fig. 1 and Table S1; abbreviations are given in Figs. 1 and 2.

relevant for efficient mycelial growth were assessed as a functional trait in saprobic fungi; contrary to our hypothesis, though, efficient growth strategies were not correlated with the fast spectrum (Fig. 3b). Instead, this axis was related to efficient biomass production and high carbon-use efficiency (CUE). And finally, the third axis (RC3 C acquisition) primarily covered C acquisition traits (Fig. 3c), including enzymatic capacities and the ability to use more complex C sources, which also positively correlated with litter decomposition ability[51].

Since these first three axes were statistically significant and together defined the main functional trait space of fungal isolates tested here, we believe they provide robust support to define the three-dimensional fungal economics space in our set of fungi. Fungal isolates were evenly distributed throughout this trait space without forming separate functional groups (Fig. 3, as found also in previous fungal trait studies[40,52]), which supports the idea of continuous spectra rather than distinct life-history categories.

### The significance of functional trait axes for soil processes

Even though our data clearly confirm distinct continuous axes forming an economics space present in soil saprobic fungi, it is still interesting to note that these main axes correlate to life-history strategies proposed by classical life-history frameworks like C-S-R, or especially its counterpart for soil microbes, the Y-A-S (yield-acquisition-stress

tolerance)[6,12]. The first axis (RC1 dense-fast) observed here involves stress tolerance (S), the second axis (RC2 flexibility) yield (Y), and the third axis (RC3 C acquisition) acquisition traits (A). However, contrary to the basic idea of these trait frameworks, described strategies did not form trade-offs, but were found on orthogonal (independent) trait axes. This result contradicts some basic assumptions underlying currently discussed life-history frameworks in soil[14]: A lack of direct trade-offs among these strategies (especially C cycling traits with stress responses) would reduce its predictive value in trait applications (see discussion about resilience below).

Classical life-history frameworks are particularly relevant in soil, because they directly link microbial traits to soil processes. Similarly, each functional trait axis described here corresponds to relevant soil processes (Fig. 3), for example melanin contents and soil aggregation ability loaded on the dense side of the first axis (RC1 dense-fast). Melanin contents reduce decomposition rates of fungal necromass, and may play a significant role in soil organic C sequestration[53], while soil aggregation is integral for soil health and C stabilization[54]. The second axis of the fungal economics space (RC2 flexibility) was mainly characterized by flexible mycelia (in terms of C:N ratios, but also explorative growth) and internal resource recycling capacities. These traits correlated with higher yields—fungal biomass and also CUE (Fig. 3)—likely caused by efficient growth due to internal

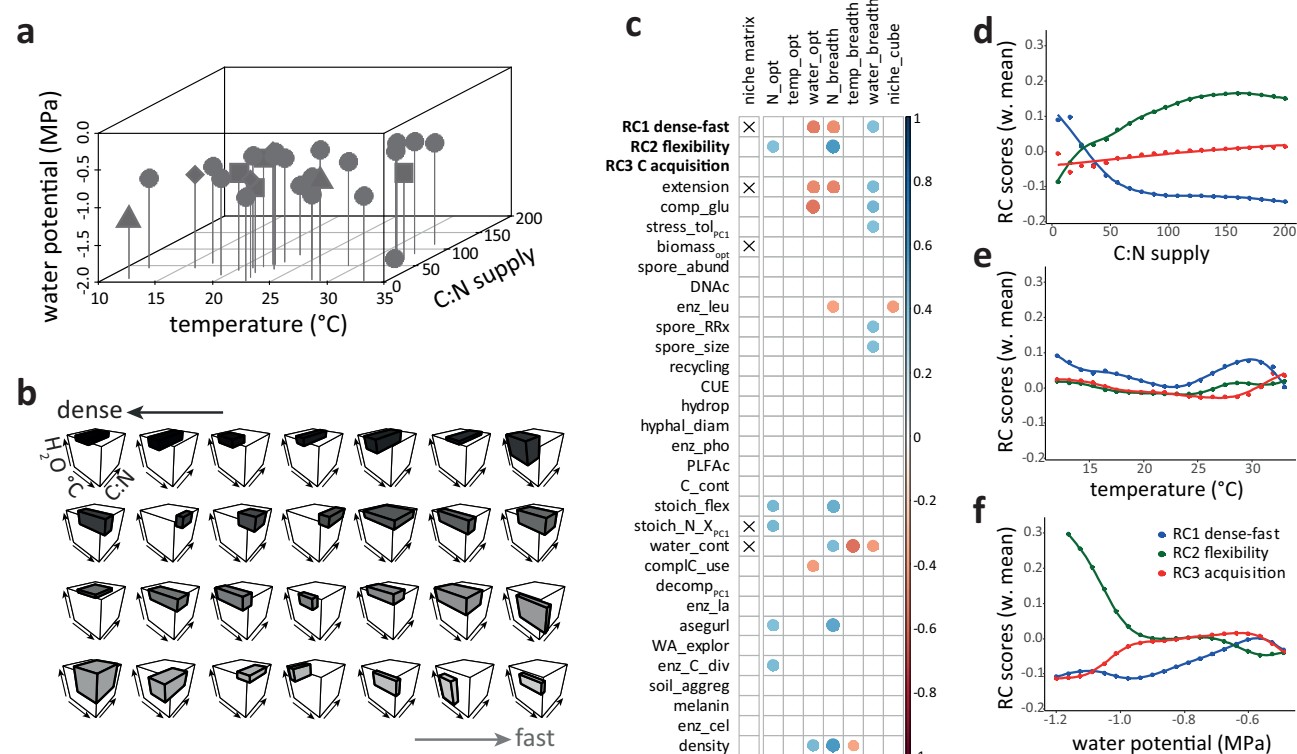

**Fig. 4 | Visualization of the fungal fundamental niches and their correlations with functional traits. a** Position of fungal optimal growth within the three-dimensional fundamental niche space. Dots represent individual isolates, shapes reflect phylogenetic placement (round: Ascomycota, square: Basidiomycota, triangle: Mucoromycota, diamond: Mortierellomycota). **b** Niche breadth of individual fungal isolates in the three-dimensional fundamental niche space. Edge lengths of the boxes correspond to respective niche breadths within each fundamental niche gradient, defined as >25% of maximum growth. Fungal isolates (individual graphs) are sorted by the scores of isolates on RC1 (rotated component 1, Fig. 3b), dark colors correspond to low values (dense mycelial growth), light gray colors to high values (fast mycelial growth). **c** Direct correlations of RC axes and individual functional traits with niche traits. Dot size and color correspond to correlation coefficients r obtained by Pearson´s or Spearman rank correlations—only significant values are displayed (P < 0.05). Crosses indicate significant predictors for

the position of fungal optima within the three-dimensional fundamental niche matrix (permutation multivariate analyses of variances (RC axes) and stepwise model selection based on redundancy analyses (individual functional traits)). Effects of fundamental niche gradients on the relative distribution of RC scores, calculated as arithmetic means of RC scores weighted by the relative abundance of fungal isolates along the fundamental niche gradients of C:N supply (**d**), temperature (**e**), and water availability (**f**) (Fig. S7). Lines represent outputs from generalized additive models. Water potential was modeled only to −1.2 MPa, below this value data were driven by two isolates only. For trait abbreviations see Figs. 1 and 2; niche matrix: optima in the three-dimensional niche matrix; N_opt, temp_opt, water_opt: isolate-specific optima in C:N (N), temperature (temp) and water availability gradients; N_breadth, temp_breadth, water_breadth: isolate-specific niche breadth in respective gradients; niche_cube: three-dimensional niche space for individual isolates (Table S1).

recycling[42], but also by a potentially larger subsequent fraction of inactive/dead C-enriched hyphae that increase CUE values[55,56]. Regarding C acquisition/mineralization traits, we anticipated that they would be captured with the dense-fast spectrum–showing trade-offs with stress tolerance or fast growing early successional traits[52,57]. However, even though slow/dense isolates seem to have high enzymatic capacity (Figs. 2, 3a), this does not necessarily translate to higher mineralization rates. Litter decomposition only correlated with the third axis of the fungal economics space (RC3 C acquisition, Fig. 3c), best predicted by the ability of fungi to use more complex C sources[51].

Stress tolerance traits are supposed to directly link soil process related traits to environmental change. We found stress tolerance to be associated with the fast (explorative) side of the first axis (RC1 dense-fast). Ecologically, the rapid colonization of new resource patches by fast (early successional) fungi exposes them to a wide range of environmental conditions. Being tolerant to environmental stressors and having wider niche breadths (Fig. 4c) will be beneficial for this strategy. Across other organism groups stress tolerance has also been associated with the fast side of the economics spectrum[58–60]. However, we here studied stress tolerance defined as the ability to maintain growth under sublethal conditions/moderate stress. Longevity and survival of hyphae are likely captured by the slow/dense side of the spectrum–if fungi follow the universal pattern found in other organism groups that slow growth correlates with longevity[48,60]. Indeed, high melanin contents (costly hyphae, Fig. 3, S1) on the dense side of the gradient support this hypothesis: Melanin is a complex biomolecule relevant for survival in extreme habitats[61]. Analyses of melanin contents in a large collection of basidiomycete fungi by ref. 62 confirmed these patterns: A similar trade-off in melanin production with fungal growth rates as well as with genes associated with stress tolerance was observed.

## Fungal trait distribution in fundamental niche spaces

To further explore the ecological significance of the fungal economics space, we correlated functional trait axes with the fundamental niche position and breadth of individual isolates. Niche breadth and tolerance to suboptimal growth conditions have been previously used as a surrogate for stress tolerance[47], but more importantly provide insights into potential functional shifts in fungal communities to changing environmental conditions[20]. Beside certain individual functional traits, the first functional trait axis (RC1 dense-fast) of the fungal economics space significantly correlated to fundamental niche optima and breadths (Fig. 4c). The position of fungal isolates along the first axis of the fungal economics space not only correlated with their position in the fundamental niche space, but also with a fungal optimum at lower water potentials (dryer conditions), associated with wider niche breadths in this gradient. By contrast, fast-growing fungal isolates showed a narrower niche breadth in respect to nitrogen availability (Fig. 4c). The second axis (RC2 flexibility) was positively correlated with optima at low N supply and wide niche breadths along the C:N gradient, while the third axis (RC3 C acquisition) showed no significant correlation with any niche traits (Fig. 4c).

To effectively describe the correlation of trait axes to fundamental niche gradients, we modeled the relative abundances of individual isolates at different fundamental niche positions (Fig. S7) and calculated respective weighted average RC scores along fundamental niche axes (isolate scores on the three RC axes of the fungal economics space (Fig. 3) under different conditions (Fig. 4d–f)). This modeling approach revealed that the relative abundance of isolates with positive scores on the dense-fast spectrum was greater under high N supply (low C:N), and slightly increased at temperatures deviating from the optimum (average fungal optimum at 25 °C (Fig. S7)). Noteworthy, scores on the second axis (RC2 flexibility) most strongly shifted along fundamental niche (resource) gradients: Fungal isolates with high scores on RC2

showed relatively less biomass reduction under low resource supply, i.e., high C:N and low water potential. Flexible mycelial strategies thus sustained growth under nitrogen and water limitations. By contrast, the scores indicative of C acquisition traits (RC3) responded least to fundamental niche gradients.

## Proposed classification of saprobic fungi in terms of life-history strategies

Even though the traits involved in the fungal economics space are specific to saprobic filamentous fungi, it does mirror life history strategies established in other organism groups. In summary, the primary axis of the fungal economics space represents a trade-off between the strategy of fast growing fungi competitive with simple sugars, versus slow growing, potentially long lived exploitative fungi with the ability to use more complex C sources. The extremes of this spectrum resemble some classical functional categories like Guerilla/Phalanx or copiotrophy/oligotrophy[22,40], but the nature of the primary trait axis indeed better fits to the concept of the slow-fast or trait economics spectrum described in plants and animals[16,18,60,63]. The concept of the economics spectrum primarily describes a trade-off between the cost of structure and rate of resource return, which leads to a successional differentiation[17,18]: Slow growth, structural investment, longevity and slow but prolonged resource return (late successional) versus fast growth, short-lived and quick resource return (early successional). In plants, this reflects the interpretation of the leaf or fine root economics spectrum along a conservative-acquisitive spectrum (slow vs. fast return). In the proposed first axis of the fungal economics spectrum, a similar pattern is associated with slow growing dense fungi with higher investment to structural components (e.g., melanin) and enzyme capacities (late successional in a soil context), whereas fast growth is linked to quick resource return by the use of simple sugars, "cheap" structures (no melanin, no other evidence so far) and intense sporulation to allow rapid colonization of new patches (early successional). Such a differentiation must be understood in light of the heterogeneous nature of soil, with high spatial and temporal fluctuations leading to rapid succession of small-scale resource patches.

Previous studies describe a dominance-tolerance trade-off in soil fungi[45,47], and sort fungal species according to their competitive ability and stress tolerance into life-history strategies[13,24,29,64]. However, competitive ability measured in artificial media (comp_glu)[47] only expresses fungal dominance in the presence of simple sugars, which allows fungi to quickly colonize productive environments (early successional) and outcompete other species[22] (by fast resource uptake and/or the production of defense compounds). Under more complex C substrate conditions though, high enzymatic capacities may characterize superior competitors (late successional)[6,46]. Consequently, fungi at both ends of the dense-fast spectrum can be competitive, just at different successional stages (Fig. 5a). Similarly, stress tolerance can be widely defined as a tolerance (maintenance of biomass production) to unfavorable conditions, but is used in practice interchangeably for environmental stressors and resource limiting conditions[6,27]. While fast growing isolates seem to be more tolerant to moderate anthropogenic stressors, fungi with dense mycelium likely can survive extreme conditions, and better cope with stress caused by a lack of rich C or nitrogen sources (Fig. 4d). Mycelial flexibility (RC2), on the other hand, helps to maintain growth under nutrient or water limitations (due to the ability to recycle internal resources and flexibly adjust mycelial strategies to resource conditions). In conclusion, the primary trade-offs observed in this study seem to be driven by niche differentiation among fungal isolates in a successional context: The three-dimensional fungal economics space, and the relative importance of axes in terms of variability explained, indicate that successional trade-offs are the primary driver of the evolution of fungal life-history strategies–a universal pattern across organism groups[18]. Further trait

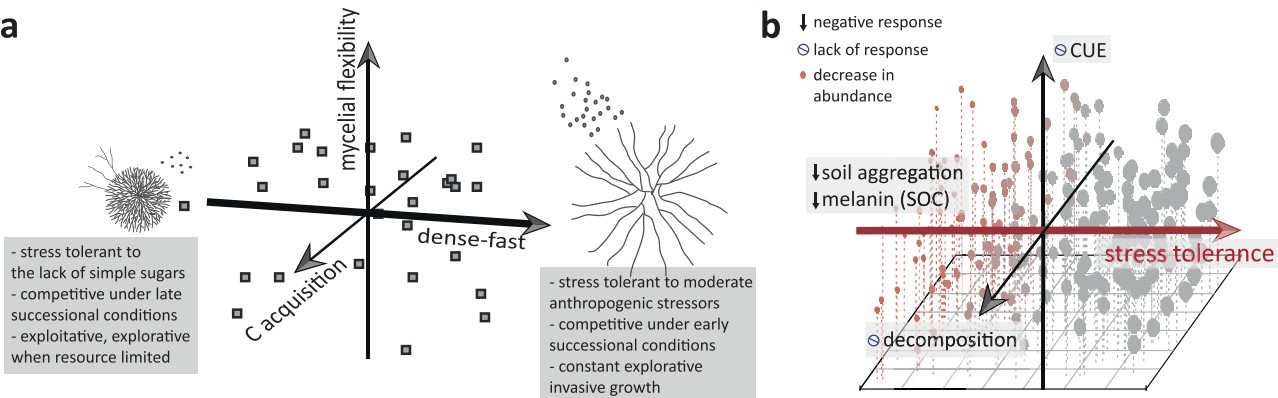

**Fig. 5 | Conceptual overview of the three-dimensional fungal economics space and its significance in an environmental context. a** The fungal economics space based on the fungal isolates tested here, with characteristics of the dense-fast spectrum displayed. Dots visualize the position of fungal isolates in the three-dimensional PCA (principal component analysis) space. Drawings show deduced mycelial structures (and spore production) at the extremes of the continuum. **b** Illustration of the proposed mechanism of functional resilience due to the orthogonal nature of main functional trait axes and their correlation to ecological strategies relevant in soil. Dots indicate a hypothetical modeled community within boundaries of the fungal economics space and the abundances of individual species/isolates in response to a moderate anthropogenic stressor (dot sizes correspond to abundances, red colors indicate negative responses to stressors). Notably, there is no shift in abundances along the second and third axis, which are independent of stress responses. Predicted responses in soil processes are indicated by arrows (round symbols = lack of response). SOC soil organic carbon, CUE carbon-use efficiency.

variability in resource or carbon use efficiency (2nd and 3rd axes) seem to be secondary factors leading to further niche differentiation and diversification in soil saprobic fungi.

## Ecological significance of the fungal economics space

The underlying assumption of models and theories addressing the role of microbial life-history strategies on C cycling rely on trade-offs among ecologically relevant traits[6,27,29]. By contrast, we observed that some major ecological strategies maintained "orthogonal" relationships with each other, which was manifested through uncorrelated PC axes (Fig. 5). We are aware that these established trait axes are based on a limited number of isolates, derived from one site only (a compromise to achieve in depth trait analyses) and future studies including more species are necessary to validate its generalizability. Still, our results indicate that variation in traits relevant for C dynamics in soil are partly uncorrelated to environmental stress responses. In our data, stress tolerance was primarily associated with the first axis. Consequently, the axes of the fungal economics spectrum representing C acquisition, as well as mycelial flexibility (mainly correlated to yield and CUE) are independent of certain stressors (Fig. 5b). No functional shifts in these parameters of C cycling would be expected despite apparent community shifts—potentially explaining the often observed functional resilience in soil[65]. Resilience in this context refers to functional stability despite microbial community shifts under environmental change, a phenomenon that in the past led to the idea of functional redundancy[65,66]. In this novel mechanism described, though, fungal isolates are not functionally redundant/similar[65] (redundancy is defined as the "ability of one microbial taxon to carry out a process at the same rate as another under the same environmental conditions"[67]). Instead, stress-tolerant fungi are as good or as bad in C cycling functions as fungi negatively affected by the stressor (Fig. 5b). This type of resilience applies to some stressors tested in this study, as well as to environmental growth conditions that shifted scores on the first axis of the fungal economics spectrum (Figs. 3, 4), and the functional traits assessed. A multitude of other relevant functions of soil fungi have not even been measured in this study; only the diversity of functional trait expressions among isolates mirrors their diverse functional roles in soil (Fig. S9c), with all of them potentially relevant in the maintenance of sustainable soils[27,68].

This high diversity and especially uniqueness of fungal isolates, in both functional traits (Fig. S9c) and fundamental niches (Fig. 4a, b), undermines the idea that losses of microbial species have no immediate functional consequences (as suggested by the concept of functional redundancy[69]). This may be the case in the short-term for specific functions under stable environmental conditions[70,71]. However, the large number of functional traits assessed here gave insights into the full diversity in functional trait spaces covered. Each isolate did not only occupy a separate position in the trait space (Fig. 3), but also showed extreme values in one or few of the traits analyzed (Fig. S9c). In parallel, isolate-specific growth optima were evenly and widely distributed throughout the fundamental niche space, and each isolate occupied its very own niche space (Fig. 4a, b). The soil environment can be seen as a heterogeneous, temporally and spatially fluctuating mosaic with seemingly endless combinations of niches, which apparently translates to a large diversity of functional and niche traits in these co-occurring fungi contributing to microbial co-existence, where each species fills a unique niche.

## Critical insights for future studies

By measuring a large and comprehensive set of traits, we were able to define a fungal economics space in tested fungal isolates and establish potential shifts in functional diversity along environmental gradients. Functional traits specific to filamentous fungal growth in heterogeneous soils defined the main three axes of the fungal economics space, namely dense versus fast growth, mycelial flexibility and C acquisition. In support of our hypotheses, the primary axis of the three-dimensional fungal economics space revealed a slow-fast spectrum in fungi similar to other organism groups, and reflected potential successional patterns of saprobic fungi growing in complex soils. Mycelial flexibility and also C acquisition traits, both discussed as being highly relevant for fungal growth and its effects on soil processes, were largely independent of this primary axis. Consequently, proposed microbial life-history strategies did not form classical trade-offs, but were found on independent orthogonal trait axes, questioning the predictive power implied by current life-history frameworks applied to soil.

To critically examine the general validity of the fungal economics space described here, we suggest testing these patterns with broader taxonomic and geographic coverage, as it has been done in the past for plants[16]. Future studies may focus on a few primary traits representative of main trait axes (Fig. 3). Following the establishment of main trade-offs in phenotypic functional traits here, further analyses may

also take advantage of -omics data[22,64]. Recent progress in metagenomics reveals a promising path to connect community composition to function, though the significance of these data in respect to phenotypic trait expressions must be further analyzed[27,59]. To keep the concepts relevant for soil biogeochemical models, the range of functional guilds included should be ecologically meaningful: We here worked with saprobic fungi that contribute to C cycling in soil to varying degrees, reflecting the functional (and phylogenetic) diversity covered in fungal sequencing studies in grasslands or agricultural soils[72]. In this set of soil fungi, the functional diversity along with fundamental niche differentiation reflects the complex dynamics acting on soil microorganisms. Unlike in classical successional litterbag or wood decomposition studies, the soil habitat is shaped by constant input and turnover of small heterogeneous resource patches coupled with small-scale variations in environmental conditions. This diversity results in rapid shifts among successional stages, and likely drives the observed trait spectrum, its continuous nature, and the unique niches of co-occurring soil fungal isolates.

To make these findings applicable to soil science, relations among functional strategies must also be resolved for bacteria[6]. This work may provide interesting starting points: Since soil bacteria are exposed to the same complex soil environment, similar patterns may be expected[28], especially since the main slow-fast spectrum appears universal across different organism groups[22,60]. Interactions among bacteria and fungi (as well as other trophic groups) will likely further influence the trait expressions and trade-offs described. This dynamic is further nuanced by the intricate complexity inherent in heterogeneous soil environments, which will be important to implement in the application of life-history frameworks in soil microbial communities. Technically, many functional traits can only be assessed in vitro, which makes this design valuable to establish physiological trait trade-offs in mycelia. It is also likely that physiologically inherent growth patterns will translate to isolate-specific dynamics in soil, especially regarding the successionally driven primary axis of the proposed fungal economics space. Still, evaluating the validity of key traits in real soil systems must be incorporated in future experimental designs, e.g., using artificial/sterile soil systems or soil-pore like microchips[73,74] in combination with metagenomic and –transcriptomic soil analyses.

In light of the complexity present in soil, the relatively clear fungal economics space we find here is still intriguing. Soil biogeochemical models are advancing rapidly and start to include microbial functional groups, making such insights into functional dynamics in microbial communities most relevant[5,6]. We hope the fungal economics space will inspire microbial ecologists to invest more into the definition of relevant functional strategies in soil, and more precisely define microbial functions that can (or cannot) be predicted based on trade-offs. These results also add an interesting mechanism of functional resilience in soil, which potentially provides a relevant buffering effect towards certain global change factors[66]. Still, the uniqueness of the functional niche space that we found also highlights potential consequences of microbial diversity loss for long-term soil sustainability.

## Methods
### Fungal isolates
An extensive trait dataset based on parallel experiments with fungal isolates of the Rillig Lab Core Set (RLCS) was combined and complemented in this study. Details on the fungal isolates and some fungal traits included here were already published elsewhere[41,75] (see Table S2 for a complete list of studies included; previously published data described below include respective references). In short, fungal strains were isolated in 2014 from soil samples taken in a natural grassland area of Northern Germany—the sampled area covered only ~100 m² (52°27′N 14°29′E). Soil washing techniques were applied to reduce the abundance of spores, and various culturing media were used to enable isolation of a diverse set of fungi[75]. Isolate identity was determined by long read sequencing (ITS1, 5.8S, ITS2 and partial LSU; sequences and cultures were deposited at NCBI (National Center for Biotechnology) and the DSMZ (German Collection of Microorganisms and Cell Cultures GmbH); accession numbers are provided in Table S3. 28 fungal isolates were selected in this study, including 19 Ascomycota, 4 Basidiomycota, 2 Mucoromycota and 3 Mortierellomycota (Table S3, Fig. S9). From the original core set (31 isolates), only three isolates within the same genus were included to achieve a more phylogenetically balanced design (Fig. S9). For maintenance, fungal isolates were kept on potato dextrose agar (PDA) at low temperatures (4° or 12 °C), and irregularly transferred to fresh medium and regenerated from stock cultures (Supplementary Note 1).

### Functional trait measurements
A multitude of functional trait variables has been measured on these RLCS fungal isolates, related to fungal growth, structure, hyphal chemical composition, resource acquisition, and mycelial growth strategies (Fig. 1; for a complete study and trait list see Table S1 and S2). Certain traits were measured repeatedly as part of different experiments, in other cases different response variables were indicative of the same fungal trait. Therefore, to reduce the amount of measured variables to meaningful traits representative of main trait axes, variables were partly reduced to individual traits by (i) calculating average values of standardized trait values (mean 0, standard deviation 1) under standard (non-stressed) conditions (Fig. S10), in case the same trait was measured repeatedly, or (ii) extracting relevant principal component (PC) axes of trait syndromes (Fig. S11). Missing trait values were imputed either by taking average values from all isolates, or, in cases where phylogenetic signals were observed, from closely related taxa[76] (only few individual data points were missing; imputed values are indicated in red in the data table available at[77]). Detailed descriptions of the methods and ecological significance of traits is given in Table S1. Table S2 includes all references and a short methodological overview of original and unpublished studies. Figure 1 provides an overview of all trait measurements and their abbreviations.

Mycelial growth. The growth of mycelia was measured by mycelial extension rate (extension; new mycelial area formed over time on Petri dishes within the linear growth phase), mycelial density (density; fungal biomass per mycelial area), and fungal biomass under standard growth conditions (biomass$_{st}$; average values of measurements under standard (non-limiting) growth conditions). Additionally, fungal biomass formed on complex C source (biomass$_{complC}$; average from biomass produced on media with xylan, cellulose, and litter[51]) as well as fungal biomass at isolate-specific optima derived from fundamental niche gradients (biomass$_{opt}$; average from biomass optima in C:N[78], temperature and water potential gradients) were included. All these final trait values represent average values of original (standardized) trait data (Table S1).

Mycelial structure and chemical composition. Several traits describing the structure or chemical composition of mycelia were measured. As such, hyphal diameter (hyphal_diam) was included as a structural trait[79]. The melanin content of mycelia was assessed using a quantitative colorimetric assay based on azure A dye (melanin). Additionally, mycelial hydrophobicity was included which relates to the presence of hydrophobins (hydrop; measured by the alcohol percentage test on mycelia[80]). Direct measurements of chemical composition were further represented by the mycelial water content (water_cont), DNA (DNAc), and PLFA concentration (PLFAc)[81] as well as element contents and ratios (stoich_C_X$_{PC1}$: PC1 of C:X (nutrient) values (positively correlated with high C:X), 43% expl. var.; stoich_N_X$_{PC1}$: PC1 of N:X values (positively correlated with high N:X), 57% expl. var.; stoich_N_X$_{PC2}$: PC2 of N:X values (positively correlated with low K values), 25% expl. var.; C_cont: mycelial C content [%]).

Enzymatic traits. Resource acquisition by enzymatic activity, mainly related to C capture, represents a most relevant trait for saprobic fungi. Here, we included enzymatic activity of cellobiohydrolase (enz_cel), laccase (enz_la), acid phosphatase (enz_pho), and leucine aminopeptidase (enz_leu) measured directly on fungal mycelia by a microplate photometric method[82]. Additionally, the number of tested C enzymes present in an isolate (enz_C_div; indicated by an API ZYM™ enzyme kit) was included[51]. Since complex C substrates may be degraded by isolate-specific and diverse enzymes, potentially not captured by simple enzyme essays, we included a more indirect assessment, i.e., the ability of isolates to grow on complex C sources (complC_use; complex carbon use ability defined by weighted averages (weighted by the degree of complexity) of fungal biomass values on glucose, cellobiose, xylan, cellulose, and litter[51]).

Sporulation. An important part of the fungal lifestyle is given by constant dispersal by asexual spore production. Thus, we included measurements of spore abundance (spore_abund; spores mg$^{-1}$ fungus), but also the shape (spore_shape; spore length/width) and size (spore_size [μm$^2$]) of asexually produced spores (except for isolates of the genus *Chaetomium* that produce only ascospores on agar media)[41].

Mycelial strategies. The mycelial growth form of fungi enables flexible strategies of exploration and exploitation of resource patches in soil. We assessed several traits representative of different strategies related to the mycelial lifestyle. First of all, the hyphal growth form allows for internal resource recycling, a trait we measured directly using a novel design (recycling; ability to produce new hyphae only with internal resources). This trait describes the ability of fungi to continue growth in the sudden absence of any external C and nutrient sources, measured by the extension rate of mycelia transferred to bare glass Petri dishes and supplied only with water. The mycelial stoichiometric flexibility in C:N ratios may also relate to internal N recycling (stoich_flex; homeostatic coefficient $1/H_{CN}$[55]), but also expresses a general flexibility in mycelial nutrient concentrations[55]. Resources are distributed spatially and temporally heterogeneous in soil. Under resource scarcity, fungi may invest into dispersal by spores (spore_RRx; relative increase in spore production under resource scarcity[41]), but also into explorative outgrowth by mycelia (asegurl; ability to switch to explorative growth under resource limitations). The latter trait was defined in this study and quantifies the observed pattern that mycelial extension rates often increase under suboptimal resource supply (Fig. S6)[78]. The preference to explore on limiting agar over glucose is another indicator of fungal exploration, showing an investment into explorative growth (WA_explor; relative mycelial extension on water agar compared to PDA on split plates[83]). Another strategic trait is represented by competitive ability in the presence of glucose, assessed as interaction outcomes on PDA (comp_glu; PC1 axis extracted from different competition scores; 90% expl. var.)[84].

Carbon-use efficiency (CUE). CUE of fungal isolates was tested with fungal isolates grown on 10% PDA, capturing all respired C during the complete growth period (CUE = biomass C/(biomass C + respired C), see Table S2).

Litter decomposition. As a proxy for actual C acquisition ability under more natural conditions, we included the decomposition rate of leaves (leaf_decomp; litter decomposition rate of *Arrhenatherum elatius* leaves) and wood (wood_decomp; decomposition rate of *Tilia cordata* wood)[51]. Both traits were combined into one litter decomposition traits (decomp$_{PC1}$) by extracting the first PCA axis (78% var. expl.). Litter samples in mesh bags were inoculated with fungal isolates over 10 weeks under laboratory conditions, and weight loss assessed[51].

Stress tolerance. As an indicator of stress tolerance, the stress response of fungal isolates (reduction in growth measured as log-response ratio) in response to environmental/anthropogenic stressors was assessed. Data on drought stress (drought_str), as well as copper (cu_str[85]) and fungicide additions (fungic_str; fungicide isopyrazam[86]) were included. To obtain a general indicator of fungal stress tolerance, PC1 of individual stress traits was extracted (stress_tol$_{PC1}$; 43% expl. var., positively correlated with stress tolerance). Since this PC1 was more strongly related to fungicide and drought stress tolerance (Fig. S11d), we also included the average value of all measurements (stress_tol$_{av}$), which presented a variable that weighted individual stressors more equally.

Soil aggregation. The (de novo) soil aggregate formation potential of fungal isolates was included (soil_aggreg[87]), since saprobic fungi play a predominant role in soil aggregation[88]. The ability of fungal isolates to form soil aggregates >1 mm in size within 6 weeks was assessed under sterile condition.

Trait variability. To assess the variability (phenotypic plasticity) of traits in our dataset, we tested the responses of trait values to varying growth conditions (C:N supply[78], temperature, C sources), and analyzed the variance explained by growth conditions versus fungal isolate identity. Detailed results and methods can be found in Supplementary Note 3. The majority of functional traits in this study were analyzed under controlled laboratory conditions. In fact, it is recommended to assess traits under optimal growth conditions, in order to allow comparisons among different species or isolates[45]. However, trait expressions also vary with environmental conditions[78], calling into question the robustness and specificity of such traits to characterize isolates or species. We tested several key traits used here beforehand, and found that even though traits changed in response to growth conditions, isolate identity remained the primary predictor of variability in trait expressions (Fig. S8). Thus, main functional traits assessed in this study are not only a product of the standard/optimal laboratory resource environment created, but indeed represent characteristic traits of fungal isolates.

## Niche traits related to the fundamental niche

The fundamental niche of fungal isolates in regards to N availability (N_opt, N_breadth), temperature (temp_opt, temp_breadth), and water availability (water_opt, water_breadth) was determined for all fungal isolates in three independent experiments (niche matrix: fungal optima in three-dimensional niche space). All isolates were grown under varying conditions of the respective environmental parameter, and mycelial biomass, density and extension rate were determined (Fig. S8d, for details see Table S1 and S2). Since extension rate and density not only reflect growth responses, but also a switch to explorative growth under suboptimal conditions[78], only biomass data were used to define isolate-specific fungal niches. N availability was modified in defined agar media[78], temperature responses based on fungal growth on PDA under varying temperature regimes, while the water potential was modified by polyethylene glycol (PEG) added to PDB (potato-dextrose broth). The gradients chosen covered a relatively wide spectrum of typical ecological ranges—C:N 5-200, temperature 12°–33 °C, water potential −0.49 to −1.91 MPa. Thus, the niche breadth was defined as ≥25% of maximum growth of each isolate under respective fundamental niche gradients (Fig. 1)[89]. Optima and niche breadth were defined through fitting skew-normal distributions (Fig. S7[89]). The complete three-dimensional niche space for individual isolates (niche_cube) was calculated assuming a cuboid niche shape, with side lengths of the respective relative niche breadths (standardized by the respective maximum niche breadth). For methodological details see Supplementary Note 2.

## Statistical analyses

All statistical analyses were conducted with R version 4.1.3[90].

The phylogenetic relatedness among isolates was determined using an alignment of the full sequence reads of ITS1, 5.8S, ITS2, and partial LSU (AlignSeqs(), DECIPHER[91]). Isolate identity was determined based on the ITS and LSU region[41]. Genetic distance was calculated (dist.ml(), phangorn[92]) and a phylogenetic tree constructed

using the unweighted pair group method with arithmetic mean (upgma(), phangorn). This tree was the basis for all phylogenetic analyses.

To define the fungal economics spectrum/space, we conducted principal component analyses (PCA) to understand the multidimensional correlation structure among functional traits[17,49,60]. All traits were checked for normality—and if needed log-transformed—and scaled to mean = 0 and SD = 1 before PCA. The significance of principal component (PC) axes and respective loadings were assessed using significance tests based on random permutations and comparisons with null distributions (PCAtest(), PCAtest[93]). This method proved to reveal more consistent results than the broken-stick method[94].

In a first attempt, a PCA including all traits was conducted, and collinear or functionally equivalent traits removed (Fig. S3). In a second step, functional traits describing only mycelial structure and chemical composition without clear ecological significance were excluded, as well as traits with ambivalent interpretations (Fig. S4). These traits may be meaningful for certain questions (Fig. 4c), but in this case they did not add additional information (further significant PC axes) to the data, and were regarded as less relevant for the interpretation of main fungal ecological strategies. The same was done with N and P uptake related enzymes, which were (negatively) co-linear with C acquisition enzymes (Fig. S4). Most importantly, removal of all these traits did not affect the general structure and arrangement of main axes (Fig. 3, S3, S4), which makes us confident to have included the most relevant functional traits in the fungal economics space.

Since final PCA trait axes did not correlate well with individual traits (Fig. S5), varimax rotation was applied on the first three significant PC axes to improve the interpretability of individual trait axes. Varimax rotation allows the rotation of PC axes in order to maximize trait loadings on resulting rotated components (RC), while maintaining the orthogonal (linearly uncorrelated) structure among different axes[49]. The phylogenetic signal present in the PCA was determined based on Pagel's $\lambda$ derived from a phylogenetic PCA, which was insignificant ($\lambda < 0.001$; phyl.pca(), phytools[95]). In addition to trait correlations by PCA, we analyzed direct correlations among functional traits using Pearson´s correlations (Fig. S1).

Despite a robust structure of the derived PCA (especially PC1 consistently reflecting the dense-fast continuum), PCA structure is inevitably affected by the insertion and removal of individual traits due to complex correlations among traits (Fig. S1). Thus, in order to explore the correlation in fungal traits by a more objective approach, we developed a PCA randomization method. In 10,000 repeats, ten fungal traits were randomly included in PCAs. PCA runs with high PC1 eigenvalues were selected (Fig. S2), and the average absolute loadings of included traits on the 1st PC axes extracted. Secondly, we explored which traits align with the main PC1 axis reflecting the dense-fast continuum. Therefore, again 10,000 random PCAs were conducted, this time including density (density) and mycelial extension rate (extension) as representative traits of the dense-fast continuum, with ten additional randomly selected traits. The average absolute loadings of traits on PC1 aligned with either extension or density were extracted and sorted accordingly (Fig. 2). PCA runs with low loadings of extension and density on PC1 were excluded from these analyses (2% of all PCAs).

As described above, niche optima and breadth (>25% of maximum growth) were determined based on skew-normal distributions fitted to fungal biomass values along fundamental niche gradients (Fig. S7). Niche traits (optima and niche breadth) were correlated with individual functional traits and RC axes (of the fungal economics space) by Pearson correlation. Only the optimum in water potential was tested by Spearman´s rank correlations due to non-normality. Correlations among the fungal optimum in the three-dimensional niche matrix (niche matrix) with individual RC axes were conducted by permutational multivariate analyses of variances (adonis2(), vegan). The relevance of individual functional traits as predictors for the trait niche matrix was evaluated based on automatic stepwise model selection of redundancy analyses (ordistep(), vegan), reducing the number of collinear variables in the resulting model testing variance inflation factors (vif.cca(), vegan). To visualize functional trait distributions along these niche gradients, the relative abundance of each isolate was calculated along each fundamental niche gradient (Fig. S7). Based on these values, the weighted arithmetic mean of isolate scores on RC axes (weighted by relative isolate abundances) were plotted against the fundamental niche gradients. Distributions were fitted by generalized additive models.

To control for phylogenetic effects on the observed findings, the main PCA was repeated using models with phylogenetic corrections (phyl.pca(), package phytools), as well including Ascomycota only (Fig. S12). The main patterns described here were also found when correcting for phylogenetic relatedness.

### Reporting summary
Further information on research design is available in the Nature Portfolio Reporting Summary linked to this article.

## Data availability
All fungal trait data analyzed in this study have been deposited in figshare under the https://doi.org/10.6084/m9.figshare.23320148. This dataset is publicly available. Cultures of fungal isolates and respective sequences were deposited at NCBI (National Center for Biotechnology) and the DSMZ (German Collection of Microorganisms and Cell Cultures GmbH). Accession numbers are available in Supplementary Information Table S3.

## Code availability
R scripts used for data analyses are publicly available under https://doi.org/10.6084/m9.figshare.23320148.

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

## Acknowledgements

T.C. acknowledges funding by the Deutsche Forschungsgemeinschaft (grant number 465123751, SPP2322 SoilSystems). M.C.R. acknowledges funding from an ERC Advanced Grant (694368). We thank Philip Hoelzmann and Manuela Scholz (Physical Geography, Freie Universität Berlin) for elemental analyses in fungal tissues, and Teresa-Magdalena Schieberlein for her support in laboratory work. We thank Will Cornwell and Ian Wright for valuable discussions contributing to the interpretation of the fungal economics spectrum.

## Author contributions

M.C.R., W.Z., A.W., D.W., S.D.V., J.R., J.R.P., L.P., Sv.M., St.M., E.L., L.L., H.H., P.G., J.G., J.d.C., D.R. A.-L., S.H., C.A.A.-T. and T.C. performed research and analyzed functional traits included in this study. M.C.R., J.R.P., J.B., M.B., A.L., S.H., C.A.A.-T. and T.C. discussed the analyses of the complete dataset and designed the first drafts of this manuscript. T.C. led the analyses and the writing of the manuscript. All authors contributed to the writing and final manuscript version.

## Funding

## Competing interests

The authors declare no competing interest.
