## [Peer Review File · Nature Communications]

Towards establishing a fungal economics spectrum in soil saprobic fungiEditorial Note: This manuscript has been previously reviewed at another journal that is not operating a transparent peer review scheme. This document only contains reviewer comments and rebuttal letters for versions considered at *Nature Communications*. Mentions of the other journal have been redacted.

Reviewer #1 (Remarks to the Author):

I reviewed the prior version of this paper submitted to [redacted] (Reviewer 1). The authors addressed most of my concerns and the revised manuscript is much improved. Overall, the paper in its current form provides useful insights into fungal ecology, particularly the apparent trade-offs between traits associated with fast and dense growth. In general, the paper is in good shape, but i do have some suggestions for further improvement.

Line 62- 75 While Grimes CRS is often invoked and discussed categorically, this framework was originally intended to represent a continuous spectrum of trade values in plant ecology. Similarly, the YAS framework is not explicitly categorical. It seems in this section the authors are trying to highlight that their proposed spectrum as especially novel because it's not a categorical framework but their portrayal of the past ecological theory is a misrepresentation.

Supplemental Figure 3. I think perhaps the axes are mislabeled here, the axes say three and four whereas the legend says two and three.

Fig 2 - I think the authors could more clearly convey in the text interpreting this figure that the inverse relationship between density and extension likely represents a physiological trade off. It would be useful to provide an explanation for why this tradeoff is likely to occur from a physiological perspective earlier in the manuscript. Also, given the importance of principal coordinates analysis for the interpretation of the results, some explanation of how to interpret axes loadings would be beneficial for entry level readers, like graduate students.

Line 233 - It could be helpful to clarify that orthogonal relationships indicate no trade-offs.

Line 261 - It's unclear what universal patterns you're referring to here, perhaps be more explicit.

Line 322 to 323 Similar to CRS, the oligotroph-copiotroph framework is not explicitly categorical and is commonly discussed as a continuum, while it has also been represented as categorical, this sentence is misleading. See below papers:

Stone, B.W.G., Dijkstra, P., Finley, B.K. et al. Life history strategies among soil bacteria—dichotomy for few, continuum for many. *ISME J* 17, 611–619 (2023). <https://doi.org/10.1038/s41396-022-01354-0>

Lauro, F. M., McDougald, D., Thomas, T., Williams, T. J., Egan, S., Rice, S., ... & Cavicchioli, R. (2009). The genomic basis of trophic strategy in marine bacteria. *Proceedings of the National Academy of Sciences*, 106(37), 15527-15533.

Figure 5. It seems like the labeling of these axes is misleading specifically “-” in front of CUE and decomposition suggests that these traits are negatively correlated with these axis. I think you intend for them to be positively associated based on Figure 3 perhaps remove the “-” or change it to a “+”.

Line 488 How many values were missing and consequently imputed?

Line 623 Please provide a citation that explains the varimax rotation or provide additional detail on exactly how this might change interpretation of the data.

Reviewer #2 (Remarks to the Author):

I reviewed a previous version of this manuscript and thought then it was an interesting and timely paper that raises some very interesting concepts applied to soil fungi.

Following an extensive revision, I am pleased to say that I found the manuscript much improved; the edits made to the text have directly addressed the issues I was concerned with and I believe the paper is much improved as a result.

I think this paper makes a useful and interesting contribution to the wider literature and it is well justified to be published in a broad-interest journal such as *Nature Communications*. I hope it inspires further experimental work to test the hypotheses/frameworks laid out here!

Reviewer #3 (Remarks to the Author):

I reviewed a previous version of this manuscript (as reviewer #3) and was quite critical for several reasons. Although some of my previous critique has been dealt with, the more fundamental problems remain, and most of them have not been brought up to proper discussion. I have focused this review on how my previous critique was handled.

1) I complained about that the work was not properly founded in previous work on fungal ecological strategies. The book chapter (Cooke and Rayner, 1984) has been added to the list of references (without any commenting text), but nothing substantial has been done to narrow the scope from a general discussion of “microorganisms” to a more strict focus on fungal biology. This is a paper about fungi, so it seem strange that fungi are first mentioned on line 99, i.e. more than halfway into the introduction!

2) I highlighted the unrealistic conditions of pure-cultures in the laboratory. While I agree with the authors that lab-based traits are not meaningless – I definitely see value in the thorough characterisation of the cultures – my impression is that fungal ecology is moving (or has already moved) away from isolates towards in situ observations in the field, e.g. by using in-growth bags combined with different -omics and other measurements on a community level. I agree that there are clear advantages with direct measurements on pure cultures, but I had expected a serious discussion about the pros & cons of “laboratory ecology”. Still, problems with artificial conditions and tissue cultures are not properly discussed.

3) I criticised the manuscript for a lack of transparency. Whereas I understand that it is difficult describe all the measurements in detail, I had hoped that the authors would have tried to be more specific about what they have actually measured and what conclusions can be drawn. Instead my feeling is that the short format and the superficial description of the “hard data” behind traits are used as an opportunity for sweeping generalisations, tailoring of the results, and overselling of conclusions. Two examples: On line 153 the word “competitive” is misleading, because here it denotes fast utilization of glucose in a growth medium (i.e. exploitation competition) while most fungal ecologists would interpret “competitive” as high capacity for interference competition (i.e. antagonistic combat for territory). Similarly, general “stress tolerance” is extrapolated from measurements on tolerance to two toxins (copper and a fungicide) and growth in a medium with low osmotic potential (PEG-addition as a proxy for drought). These multi-layered extrapolations from a few specific measurements under highly artificial conditions to broad ecological trait, discussed in a context of field conditions, are easy to miss for the reader. My impression is that the authors have not done their best to help the reader to navigate, but rather chose to “paint with a broad brush”.

4) I complained about the convoluted analyses with traits assigned to different categories in a ambiguous way. Much of these problems have been sorted out in the revised manuscript, which is now much easier to follow.

Thus, I maintain my view that the manuscript presents a nice and valuable multifactorial characterization of fungal cultures, which may (or may not) have some relevance for our understanding of fungal ecology. It may well be published, but in a more specialised journal and not as a foundation for a general trait-framework of fungi.

Specific comments:

47 To me "soil health" is a problematic buzzword, which is not well defined but invented by scientists to sell research projects to politicians and funders. I would avoid using it in a scientific text.

94 "in a first step we need to" should be changed to "as a first step we may", because there are many options whereby fungal traits can be studied directly in the field.

113 I would think saprotrophic fungi take up most nutrients in organic form (e.g. N as amino acids/sugars).

195 With the diverse stressors that may affect fungi, I would be more careful here. I don't think it is justified to generalize about stress tolerance without measuring e.g. tolerance to low pH. I would change to the more specific: "The fast side coincided with a high tolerance to stress induced by copper, fungicide or low osmotic potential".

207-208 Here the generality of the data is vastly overstated. Please change "we believe they provide robust support to define the tree-dimensional fungal economics space" to "and may be useful to describe ecophysiological properties of fungal cultures"

284 This part is impossible to understand without reading the supplement, since "relative abundance" has to relate to some kind of sampling, but it is not clear what was sampled.

340-346 Here competition is discussed generally, but in fungal ecology interference competition, i.e. territorial competition for space, is often emphasised. Here, only exploitation competition, i.e. rapid use of available resources is considered.

REVIEWER COMMENTS

Reviewer #1 (Remarks to the Author):

I reviewed the prior version of this paper submitted to [redacted] (Reviewer 1). The authors addressed most of my concerns and the revised manuscript is much improved. Overall, the paper in its current form provides useful insights into fungal ecology, particularly the apparent trade-offs between traits associated with fast and dense growth. In general, the paper is in good shape, but i do have some suggestions for further improvement.

We thank the Reviewer for the positive evaluation of our revised manuscript, and the thorough revision of it.

Line 62- 75 While Grimes CRS is often invoked and discussed categorically, this framework was originally intended to represent a continuous spectrum of trade values in plant ecology. Similarly, the YAS framework is not explicitly categorical. It seems in this section the authors are trying to highlight that their proposed spectrum as especially novel because it's not a categorical framework but their portrayal of the past ecological theory is a misrepresentation.

We thank the Reviewer for this helpful comment. From the more recent literature we indeed understood one of the main differences between the economics spectrum and CSR to be continuous vs. categorical. But we agree that this should not be the main argument to suggest an economics spectrum, since the triangles also present continuous spaces.

We clarified this argument throughout the text.

L63/64: “Frameworks like the widely applied Grime’s C-S-R (competitor-stress tolerant-ruderal) triangle expand on this simple idea and sort species into respective life history strategies^{12, 13}”

L72/73: “In practice, though, these simple categories fail to capture the continuous nature of trait expressions along independent dimensions..”

L232-234: “Even though our data clearly confirm distinct continuous axes forming an economics space present in soil saprobic fungi, it is still interesting to note that these main axes correlate to life-history strategies proposed by classical life-history frameworks like C-S-R...”

L234, 237, 240: We removed the word “categorical” when describing classical trait frameworks

Supplemental Figure 3. I think perhaps the axes are mislabeled here, the axes say three and four whereas the legend says two and three.

Thank you for the thorough reading. The figure legend was wrong, the legend now states three and four.

Fig 2 - I think the authors could more clearly convey in the text interpreting this figure that the inverse relationship between density and extension likely represents a physiological trade off. It would be useful to provide an explanation for why this tradeoff is likely to occur from a physiological perspective earlier in the manuscript. Also, given the importance of principal coordinates analysis for

the interpretation of the results, some explanation of how to interpret axes loadings would be beneficial for entry level readers, like graduate students.

Thanks for this comment! As suggested, we added a bit more physiological background on this primary dense-fast spectrum in this first part of the results section, also referring to the ecological patterns described in the introduction of fungal exploitation vs. exploration (L107-110).

L151-156: “While slow fungi were characterized by *dense* highly-branched mycelia, *fast* fungal isolates showed high mycelial extension rates, i.e., investment into rapid tip growth. This primary axis likely represents a physiological trade-off based on the simple nature of mycelial development (characterized by tip growth, branching and negative autotropism³⁹), but also reflects the fungal strategies of exploitation versus exploration in soil.”

For a more clear interpretation of PCA results, we modified the following sentence to “The correlation (PC loadings) of further functional traits with this 1st PC axis revealed the overall trait continuum..” (L156) and added a short summary of the rationale to use PCA for this dataset in the statistical methods sections “To define the fungal economics spectrum/space, we conducted principal component analyses (PCA) to understand the multidimensional correlation structure among functional traits” (L627).

Line 233 - It could be helpful to clarify that orthogonal relationships indicate no trade-offs.

Thank you for this comment. We now define orthogonal when first mentioning as “linearly uncorrelated” (L194).

Line 261 - It's unclear what universal patterns you're referring to here, perhaps be more explicit.

Thanks for pointing this out. We clarified “if fungi follow the universal patterns found in other organism groups that slow growth correlates with longevity^{48, 61}” (L267/268).

Line 322 to 323 Similar to CRS, the oligotroph-copiotroph framework is not explicitly categorical and is commonly discussed as a continuum, while it has also been represented as categorical, this sentence is misleading. See below papers:

Stone, B.W.G., Dijkstra, P., Finley, B.K. et al. Life history strategies among soil bacteria—dichotomy for few, continuum for many. ISME J 17, 611–619 (2023). <https://doi.org/10.1038/s41396-022-01354-0>

Lauro, F. M., McDougald, D., Thomas, T., Williams, T. J., Egan, S., Rice, S., ... & Cavicchioli, R. (2009). The genomic basis of trophic strategy in marine bacteria. Proceedings of the National Academy of Sciences, 106(37), 15527-15533.

We agree and removed the misleading word “continuous” from the sentence. Now we only refer to the nature of the primary trait axis to clarify the point (L329-331): “The extremes of this spectrum resemble some classical functional categories like Guerilla/Phalanx or copiotrophy/oligotrophy^{22, 40}, but the nature of the primary trait axis indeed better fits to the concept of the slow-fast or trait economics spectrum described in plants and animals^{16, 18, 61, 64}.”

Figure 5. It seems like the labeling of these axes is misleading specifically “-” in front of CUE and decomposition suggests that these traits are negatively correlated with these axis. I think you intend for them to be positively associated based on Figure 3 perhaps remove the “-” or change it to a “+”.

Thank you. We replaced the admittedly confusing “-” sign with a symbol for a lack of response “⊖”, explained in the figure legend (L379).

Line 488 How many values were missing and consequently imputed?

In fact, this only applies to very few individual values. In case of the functional traits included in the final PCA (Fig. 3) only melanin had one missing value, which was imputed by a nearby taxon and could be validated based on colours of that isolate.

We added an explanation to the methods, also referencing the dataset available at figshare: “(only few individual data points were missing; imputed values are indicated in red in the data table available at⁷⁶.” (L505-506)

Line 623 Please provide a citation that explains the varimax rotation or provide additional detail on exactly how this might change interpretation of the data.

Thank you for this comment. We added a short sentence explaining the underlying principle of varimax rotation, and added a citation (Carmona et al. 2021) in which varimax rotation was applied to a similar trait PCA with helpful explanations given in it.

L645-647: “Varimax rotation allows the rotation of PC axes in order to maximize trait loadings on resulting rotated components (RC), while maintaining the orthogonal (linearly uncorrelated) structure among different axes⁴⁹.”

Reviewer #2 (Remarks to the Author):

I reviewed a previous version of this manuscript and thought then it was an interesting and timely paper that raises some very interesting concepts applied to soil fungi.

Following an extensive revision, I am pleased to say that I found the manuscript much improved; the edits made to the text have directly addressed the issues I was concerned with and I believe the paper is much improved as a result.

I think this paper makes a useful and interesting contribution to the wider literature and it is well justified to be published in a broad-interest journal such as Nature Communications. I hope it inspires further experimental work to test the hypotheses/frameworks laid out here!

We are very grateful for the positive evaluation of our study, and we want to thank the Reviewer for the great contribution during the Reviewing process! We found the comments very helpful for revising the text.

Reviewer #3 (Remarks to the Author):

I reviewed a previous version of this manuscript (as reviewer #3) and was quite critical for several reasons. Although some of my previous critique has been dealt with, the more fundamental problems remain, and most of them have not been brought up to proper discussion. I have focused this review on how my previous critique was handled.

We thank the Reviewer for a second round of thorough revision! We implemented the suggestions and mark certain limitations in the text more clearly, also following suggestions of the Editor.

1) I complained about that the work was not properly founded in previous work on fungal ecological strategies. The book chapter (Cooke and Rayner, 1984) has been added to the list of references (without any commenting text), but nothing substantial has been done to narrow the scope from a general discussion of “microorganisms” to a more strict focus on fungal biology. This is a paper about fungi, so it seem strange that fungi are first mentioned on line 99, i.e. more than halfway into the introduction!

We submitted this manuscript to a Journal with broader scope, because we are convinced these results are relevant not only to this field but to soil (microbial) ecologists in general. The patterns described are specific to fungi, but the implications of our findings on the interpretation and application of life-history frameworks to soil microbial communities are relevant for soil studies in general. For this reason, the introductory text focuses more strongly on broader ecological concepts, rather than fungal ecology per se. Nevertheless, the title contains the word fungi, and the abstract mentions fungi very early, so it is unlikely that readers are misled.

2) I highlighted the unrealistic conditions of pure-cultures in the laboratory. While I agree with the authors that lab-based traits are not meaningless – I definitely see value in the thorough characterisation of the cultures – my impression is that fungal ecology is moving (or has already moved) away from isolates towards in situ observations in the field, e.g. by using in-growth bags combined with different -omics and other measurements on a community level. I agree that there are clear advantages with direct measurements on pure cultures, but I had expected a serious discussion about the pros & cons of “laboratory ecology”. Still, problems with artificial conditions and tissue cultures are not properly discussed.

Thank you. We now made a very clear point early on in the abstract and Introduction that trait data presented in this study were assessed *in vitro*. As a consequence of this limitation we also marked more clearly that this is a first attempt (see title and L126) to describe a fungal economics spectrum. We agree that this must be evaluated in future studies under more natural conditions, and with more isolates at broader scales. We added now a more thorough discussion on this point in the Conclusions section (L457-463):

“Technically, many functional traits can only be assessed *in vitro*, which makes this design valuable to establish physiological trait trade-offs in mycelia. It is also likely that physiologically inherent growth patterns will translate to isolate-specific dynamics in soil, especially regarding the successional driven primary axis of the proposed fungal economics space. Still, evaluating the validity of key traits

in *real* soil systems must be incorporated in future experimental designs, e.g., using artificial/sterile soil systems or soil-pore like microchips^{74, 75} in combination with metagenomic and –transcriptomic field studies.”

We still think this is a very useful dataset to understand potential functional trade-offs in fungal trait expressions, and filter out some relevant traits. There are techniques to assess fungal growth traits *in vivo*, but all of them still come with limitations when analysing traits of saprobic soil fungi (this may be different in wood decomposers). And for many of the traits discussed here there would still not be a method to assess them in soil, especially not at relevant mycelial scales. E.g., element contents in individual hyphae can be assessed by spectroscopic techniques, but only in a very limited range of hyphal pieces, where we know there is high variability due to internal recycling. In-growth bags may be helpful to filter out fungi from complex soil environments, but depending on the conditions not all fungi will colonize the bags. –omics data will be very useful in future studies, but their relevance for actual phenotypic expressions of traits is not resolved, yet, and for many relevant traits it is not clear which genes are indicative of certain life-history strategies, a point also further clarified in the Conclusions (L438/439).

3) I criticised the manuscript for a lack of transparency. Whereas I understand that it is difficult describe all the measurements in detail, I had hoped that the authors would have tried to be more specific about what they have actually measured and what conclusions can be drawn. Instead my feeling is that the short format and the superficial description of the “hard data” behind traits are used as an opportunity for sweeping generalisations, tailoring of the results, and overselling of conclusions. Two examples: On line 153 the word “competitive” is misleading, because here it denotes fast utilization of glucose in a growth medium (i.e. exploitation competition) while most fungal ecologists would interpret “competitive” as high capacity for interference competition (i.e. antagonistic combat for territory). Similarly, general “stress tolerance” is extrapolated from measurements on tolerance to two toxins (copper and a fungicide) and growth in a medium with low osmotic potential (PEG-addition as a proxy for drought). These multi-layered extrapolations from a few specific measurements under highly artificial conditions to broad ecological trait, discussed in a context of field conditions, are easy to miss for the reader. My impression is that the authors have not done their best to help the reader to navigate, but rather chose to “paint with a broad brush”.

Following the suggestions made by the Reviewers and Editor we marked more clearly throughout the manuscript (title, abstract, Introduction, Methods and Conclusions) that this dataset has certain limitations and must be seen as a “first attempt” towards establishing a fungal economics spectrum.

The point raised by the Reviewer on the definition of competition and stress tolerance is discussed in detail in L324ff.

We agree that it is quite challenging to summarize such a large dataset with relatively limited numbers of words. This is a limitation to many valuable big datasets in ecology. We hope to discuss some specific points in more detail in future manuscripts.

4) I complained about the convoluted analyses with traits assigned to different categories in a ambiguous way. Much of these problems have been sorted out in the revised manuscript, which is now much easier to follow.

Thank you.

Thus, I maintain my view that the manuscript presents a nice and valuable multifactorial characterization of fungal cultures, which may (or may not) have some relevance for our understanding of fungal ecology. It may well be published, but in a more specialised journal and not as a foundation for a general trait-framework of fungi.

Specific comments:

47 To me "soil health" is a problematic buzzword, which is not well defined but invented by scientists to sell research projects to politicians and funders. I would avoid using it in a scientific text.

We exchanged it with "soil sustainability" (L51, 405)

94 "in a first step we need to" should be changed to "as a first step we may", because there are many options whereby fungal traits can be studied directly in the field.

L95 We exchanged it as suggested

113 I would think saprotrophic fungi take up most nutrients in organic form (e.g. N as amino acids/sugars).

L113 we removed the word "inorganic"

195 With the diverse stressors that may affect fungi, I would be more careful here. I don't think it is justified to generalize about stress tolerance without measuring e.g. tolerance to low pH. I would change to the more specific: "The fast side coincided with a high tolerance to stress induced by copper, fungicide or low osmotic potential".

We agree there are some other relevant stressors which should be evaluated in future studies. Still, including three stressors is by far exceeding any previous attempts to assess "stress". The stressors we here refer to are given in brackets (L202).

207-208 Here the generality of the data is vastly overstated. Please change "we believe they provide robust support to define the tree-dimensional fungal economics space" to "and may be useful to describe ecophysiological properties of fungal cultures"

Thank you. We changed it to "we believe they provide robust support to define the three-dimensional fungal economics space in our set of fungi". (L212/213)

284 This part is impossible to understand without reading the supplement, since "relative abundance" has to relate to some kind of sampling, but it is not clear what was sampled.

Thank you for pointing this out. We clarified: "we modeled the relative abundances of individual isolates at different fundamental niche positions" (L289/290)

340-346 Here competition is discussed generally, but in fungal ecology interference competition, i.e. territorial competition for space, is often emphasised. Here, only exploitation competition, i.e. rapid use of available resources is considered.

Thank you for this comment. In fact, in these competition experiments both processes are involved. Some fungal isolates clearly outcompete by faster resource use, while others produced defense compounds to keep other fungi at distance or push back. We clarified that this is a mix of both processes “which allows fungi to quickly colonize productive environments (early successional) and outcompete other species²² (by fast resource uptake and/or the production of defense compounds)” (L348-350)

The territoriality the reviewer refers to is in fact very often seen in studies of basidiomycetes, which also dominate the literature; however, we work with a much more phylogenetically diverse set of fungi, and rarely see strong territoriality.